# Study examining the significant role of intellectual property protection in driving radical technological innovation among national research project teams, employing PLS-SEM and ANN modeling

**Wei Chen[1], Jianhui Yin[2]\*, Ye Tian[2], Haixu Shang[1], Yuan Li[2]**

**1** School of Management, East University of Heilongjiang, Harbin, China, **2** School of Economics and International Trade, East University of Heilongjiang, Harbin, China

\* 23002005hljeu.edu.cn

## Abstract

This study examines the role of intellectual property protection (IPP) in enhancing radical technological innovation (RTI) within national research project teams, using an innovation-driven theory and an ability-motivation-opportunity (AMO) perspective. This study utilizes a sample of 336 national research project team members from various Chinese universities, research institutes, and corporations to analyze the theoretical model. Additionally, a two-stage hybrid partial least squares structural equation modeling (PLS-SEM) approach, combined with artificial neural network techniques (ANN), is employed to evaluate the hypotheses. The empirical findings of this study reveal a positive association between the intensity of IPP and RTI within national research project teams. Research and development investment intensity (R&DII) is identified as the primary predictor, while integrated leadership (IL) and group potential (GP) play crucial moderating roles. These groundbreaking findings extend the scope of innovation-driven and AMO theories, providing a proactive model for national research project teams to propose improvements to the IPP system, ultimately enhancing the realization of RTI.

**Data Availability Statement:** All relevant data are within the paper and its Supporting information files.

## 1. Introduction

Innovation is the impetus behind worldwide development progressions. National research programs are developed to promote innovative development and growth within specific domains, aiming to address intricate, interdisciplinary, and multifaceted scientific issues. The scientific research project teams in China can be classified into three levels: those financed by national special funds, those supported by the Chinese Academy of Sciences, the National Defence Science and Industry Commission, and provinces and municipalities directly under the Central Government; those supported by some universities, research institutes, and enterprises; and those financed by some universities, research institutes, and enterprises on their own. The novelty of the innovation determines whether it will be incremental or radical [1]. RTI is a complex, high-risk, and lengthy process that cannot be achieved without a

**Funding:** This work was supported in part by the Key projects for economic and social development in Heilongjiang Province of China under Grant 23301, in part by the Harbin Science and Technology Bureau Science and Technology Programme Projects of China under Grant ZC2023ZJ014007, and in part by the Heilongjiang Oriental College Research and Innovation Team Building Project of China under Grant HDFKYTD202108.

**Competing interests:** The authors have declared that no competing interests exist.

combination of internal and external factors, such as financial support, talent pool, effective team management, favorable economic environment, and policy support. For instance, studies have shown that the digital economy significantly enhances a company's RTI capabilities by alleviating financing constraints, facilitating talent aggregation, and increasing environmental uncertainty [2].

Nonetheless, national research project teams often face difficulties in implementing RTI. The development of an appropriate organizational structure necessitates skilled professionals, creative thinking, and adequate financial backing. This, in turn, fosters efficiency and multiplies innovative opportunities. Inversely, handling intellectual property rights disputes can be particularly intricate when multiple stakeholders or organizations are involved, which may lead to disagreements among project members and impede data sharing, transfer, and future collaborative endeavors [3]. Relevant scholars have examined the significant positive impact of IPP on the technological innovation within national research project teams [4]. Nonetheless, some scholars argue that strict IPPs may negatively affect technological innovation in national research projects [5]. Therefore, further exploration is necessary to understand the impact of IPP on RTI. Currently, there is a lack of literature examining the role of IPP from the perspectives of IL, innovation milieu (IM), R&DII, and GP, in addition to field research.

This study sampled members from national research project teams affiliated with universities, research institutes, and enterprises to compile a corresponding dataset. The PLS-SEM and ANN techniques were utilized for empirical analysis. According to the PLS-SEM results, the moderate strengthening of IPPs may help national research project teams to actively explore innovations and protect their intellectual assets, promote the rational sharing and transfer of innovations, and help research teams to improve their competitive advantage, thus contributing to the realisation of RTI. Through ANN- conducted sensitivity analyses, R&DII has been identified as the most crucial predictor. This provides substantial financial backing for research teams to attain groundbreaking results, and assists researchers in focusing their efforts and resources, thereby streamlining the critical preliminary research and trial phases.

The primary contributions of this research can be condensed into three key areas: ①Expanding two fundamental theories: the innovation-driven theory and the AMO theory. These theories enhance the existing theoretical framework for RTI studies on IPP and national research project teams. Research has been conducted to explore the influence of external factors on the RTI of national research project teams, with a focus on IPPs [6], instead, this paper highlights the importance of internal team elements, such as IL, IM, R&DII, and GP, in driving technological advancements. This study delves into the Role of RTI, a subject that has not been extensively examined in the context of technological innovation, thereby enriching the existing literature in this field. ②The objective of this passage is to enhance the research framework for national research project team management, which is based on the interplay between IPP, IL, IM, R&DII, GP and research team RTI. Existing research predominantly focuses on single latitude analysis [7]. This study fills the gaps in the literature on RTI brought about by the national research project team in multi-dimensional time, thereby contributing to the growth of research in this area.③This paper discusses the policy implications and recommends that countries establish a robust intellectual property policy to protect the rights and interests of scientific research outcomes. Furthermore, the development of policies that foster collaboration between scientific research teams and industry for technology transfer is crucial. Governments and organizations can enhance their leaders' integrated capabilities through effective training and development programs. Moreover, providing increased financial support for scientific research projects can lead to increased R&D investments, thereby increasing the likelihood of RTI within these research teams.

## 2. Theoretical frameworks and hypotheses

To better elucidate the influence between IPP and RTI in national research programs, this study utilizes Michael Eugene Porter's innovation-driven theory and Appelbaum's Ability-Motivation-Opportunity (AMO) theory to examine how IPP can foster breakthroughs in technological innovations within research teams [8, 9]. Porter's theory delineates four pivotal stages of national economic development: factor-driven, investment-driven, innovation-driven, and wealth-driven. It emphasizes that the innovation-driven stage enhances innovation levels by leveraging technology, knowledge, systems, and management as primary drivers, thus optimizing resource allocation, enhancing labor quality, and improving the research environment [10]. IPP enhances firm productivity by stimulating technological innovation within research teams, improving investment efficiency, and bolstering competitive advantage [11]. Barrios et al. highlight that the interplay between open innovation and the use of non-disclosure agreements for protecting intellectual property rights can influence the innovation performance of manufacturing and service firms [12]. Conversely, AMO theory suggests that satisfying team members' abilities provides them opportunities to optimally utilize their skills. By meeting the ability, motivation, and opportunity needs of team members, maximal team benefits are realized. These frameworks were chosen for their robust analytical structure, enabling a deep understanding of research innovation complexities, particularly through the lens of institutional factors and team dynamics. Innovation-driven theory underscores the essential role of knowledge and technology in economic growth, highlighting the critical importance of intellectual property regimes in sustaining innovation [13]. Meanwhile, AMO theory offers a micro-level perspective that provides a practical framework for analyzing and influencing team dynamics, detailing how competencies, incentives, and opportunities interact at the team level to impact organizational performance [14].

This study aims to address a critical gap in existing research, which broadly acknowledges that IPP stimulates innovation but often lacks comprehensive analysis of its effects on the innovation processes within research teams. Utilizing the analytical framework of the Ability-Motivation-Opportunity (AMO) theory, this paper investigates how IPP fosters innovation by enhancing researchers' motivation (e.g., through patent incentives and increased R&D funding), empowering them via training and technical support, and creating opportunities for high-quality innovation outcomes (e.g., by promoting cross-disciplinary collaborations and enhancing integrative leadership). For instance, IPP increases researchers' incentives to engage in high-risk projects by ensuring commercial returns on research results; it strengthens research teams' access to critical technological resources through institutional mechanisms [15]; and it facilitates cross-disciplinary collaboration that enhances team communication and cooperation [16]. This integration of theory not only deepens our understanding of IPP's role in research teams but also offers actionable insights for enhancing research management and policy-making, thus providing substantial theoretical and practical contributions to the literature. By examining the multifaceted impacts of IPP, this study enriches the existing literature and lays a scientific foundation for developing more effective policies to promote innovation.

The success of national research programs in achieving RTI and their impact on RTI are primarily influenced by R&D investments and the stock of intellectual capital. Research indicates that IPP offer both positive external incentives and negative external challenges for RTI. A higher R&D investment creates a more favourable environment for promoting technological innovation and progress. Enhanced IPP allows national research program teams to financially benefit from RTI, thereby enabling further investment in research and development (R&D) activities [17]. Drawing on Clayton Christensen's The Innovator's Dilemma, many organizations are drawn towards maximizing profits and meeting current demands, yet overlook the

exploration and development of innovations in niche technology tracks [18]. IPP enhance the exclusivity of knowledge through legal measures, which amplifies the monopoly power of innovators. This elevation in monopoly power significantly increases the discounted value of expected profit streams from innovations, thereby directly stimulating innovation incentives [19–22]. IPP enhances green innovation in corporate research teams by mitigating R&D spillover losses and easing financing constraints [23]. RTI enables the transformation of knowledge and technology into innovative outcomes within national research programs, fostering and expanding the boundaries of the discipline and its applications across various fields. Consequently, proactive advancements in technological innovation are crucial for national research programs to prevent losses in science and technology competitiveness, not just for individual enterprises but also for the nation as a whole. Furthermore, a robust inventory of knowledge assets serves to bolster technological innovation capabilities. A substantial knowledge asset inventory will foster a stronger technological foundation for innovation teams working on national research projects. A shortage of such assets could hinder the team's progress, establish a lower starting point for technological innovations, and heighten the challenges in achieving breakthroughs. National research project teams transcend organizational boundaries to harness external technological resources and knowledge, thereby accelerating internal innovation, enhancing inter-organizational collaboration, and fostering RTI [24–27]. Through technological development, organizations integrate externally acquired technologies with internal resources and capabilities, generating new knowledge that benefits partners and catalyzes future collaborations [26, 28]. Conversely, the IPP system, by establishing a legally protected monopoly, restricts knowledge spillovers and complicates the accumulation of knowledge, thus impeding cumulative innovation [29, 30]. Moreover, IPPs strengthen the monopoly power of IPP owners and diminish the incentives for open competition and R&D investment, ultimately discouraging innovation [31, 32]. The theory of knowledge creation suggests that the sharing of tacit knowledge within an organization, combined with the processes of socialization, externalization, integration, and internalization, can lead to knowledge innovation [33]. For instance, a discord among team members, like in an innovation team, might cause the inadequate utilization of knowledge assets, impeding the achievement of RTI in national research projects [34].

The full utilization of knowledge assets and substantial R&D investments is crucial for enhancing the realization of RTI. Nevertheless, pure market forces may demonstrate some market failures [35]. It was recognized that additional policy measures are required to enhance innovation capacity. The government-promoted IPP system plays a vital role in fostering innovation improvement. By effectively protecting proprietary rights and reducing technological innovation uncertainties, IPP contribute to achieving RTI in national research projects [36]. The IPP can implicitly influence innovation team leaders by mitigating their apprehensions about innovation disclosure, thereby encouraging them to adopt more proactive technological innovation behaviors. Grimaldi et al. assert that no IPP strategy significantly impedes the success of RTI [37]. Inversely, either insufficient or exaggerated IPP safeguards in research projects can significantly affect R&DII [38], and in turn, impact RTI. Hagedoorn and Zobel observe that many firms in developed countries across Europe and North America regard formal contractual agreements as crucial legal mechanisms for safeguarding intellectual property rights in collaborations with Open Innovation partners [39]. The spillover effect tends to be weak when IPP awareness is heightened, while the complexity of the technology makes it difficult to replicate knowledge assets during the transfer process. As a result, it's more plausible for imitators to avoid infringing on the originator [3], thus fostering RTI. Based on this, this paper proposes:

**H1**: IPP significantly enhances RTI within national research project teams.

To a certain extent, the strength of IPP's influence on team leaders also leads to significant variations in IL. Specifically, the growing influence of IPP has promoted a collaborative decision-making and sharing characteristic among IL within national research project teams. The current IPP regime is theoretically justified largely on moral grounds derived from utilitarianism, and theories of labor and personhood, which emphasize private ownership and the maximization of economic profits [40]. This foundation aligns consistently with systemic cooperation in RTI, particularly regarding innovation goals that encompass economic, cultural, social, and environmental sustainability dimensions. It also underscores the necessity of reciprocity and collective efforts to achieve co-specialization in shared technologies [41]. For instance, RTI relies not only on sharing IPP innovations and technologies but also on diversifying and integrating knowledge among leaders and stakeholders to manage conflicts of interest and maintain productive cooperation that leverages the potential of IP [42]. This can be attributed to the environment of intense competition, which impedes the achievement of RTI. The team's integrative leaders direct their followers' behavior consider whether there is a risk of disclosure of the security aspects of technological innovation information [43]. Specifically, it proposes that strengthening IPP could lead to more collaborative decision-making, thereby reducing the likelihood of such information being disclosed within national research project teams. This, in turn, has a significant positive impact on the development of IL.

Variations in the intensity of IL across national research projects can lead to significant differences in the Realization of RTI. As Crosby and Bryson suggest, IL is primarily centered on multiple organizations or groups to achieve public value [44]. This leadership style represents a new paradigm emerging from the integration of leadership and innovation, characterized by integrative leaders who possess distinct qualities that allow them to identify and leverage environmental opportunities [45]. It is posited that IL is intrinsically linked to opportunity identification [46]. A fundamental aspect of entrepreneurial leadership involves embracing the concept of technological innovation, which connects breakthrough opportunities with potential markets [47, 48]. The success of RTI hinges on the mindset and culture of leaders and actors, who must maintain a dynamic approach to opportunities and possess the requisite skills for their interpretation [49–51]. The technological innovation of a team is influenced by the individual's knowledge, skills, and motivation to innovate in their respective field [52]. IL significantly contributes to the development of RTI by facilitating these processes. Numerous studies have demonstrated that IL is a significant motivator in RTI [53–57]. In a national research project, a stronger IL of a team enhances the utilization of the team organization's aspects in planning and decision-making, transcends organizational boundaries, and generates value towards achieving RTI. Tsai et al. suggest that IL boosts innovative expectations and satisfaction, leading to increased intra-workgroup competition and knowledge sharing [58]. Nonetheless, IL bolster team resource allocation and selection, providing organizations with enhanced flexibility, responsiveness, and risk management capabilities, thus making a substantial contribution to achieving RTI within national research project teams [44].

IL also significantly contributes to the success of national research projects by mediating the impact of IPP strength. This is attributed to the fact that IPP provides a conducive institutional environment for team leaders to effectively manage multi-channel resource integration, which in turn promotes product development and commercialization [59]. National research projects necessitate the integration of more internal and external resources. Integrative leaders significantly influence subordinates' work-related behaviors and attitudes within the workplace [60], and they critically impact self-efficacy and creativity [61, 62]. IL, which is grounded in the integration of elements and strategic objectives, aims to enhance relationship stickiness.

By influencing IPP, team leaders integrate multi-faceted resources, including technical skills, decision-making, teams, and mechanisms. Mutual respect and service, fundamental to an IL approach, enhance innovation self-efficacy, which in turn boosts innovation effectiveness [63]. This method is more conducive to the national research project team in achieving RTI. This paper posits the following:

**H2a:** IPP has a significant positive effect on IL of national research project teams.

**H2b:** IL of national research project teams has a significant positive effect on RTI.

**H2c:** IL of national research project teams has a significant mediating effect on the impact of IPP and RTI.

The strength of IPP may, to some extent, boost team IM within national research projects. In particular, stringent IPP laws serve to deter plagiarism, and IP licensing provides a means to utilize others' patented technologies, which is more conducive to the development of good team IM. IPP can indirectly influence the IM within teams, impacting trust both positively and negatively. The economic theory of cost transactions posits that trust acts as a substitute for contracts [64]. An increased reliance on contracts might diminish trust, as contracts may signal a lack of trust and the anticipation of opportunistic behavior, subsequently fostering an unfavorable IM [65–67]. Kadefors discovered that detailed contract specifications and stringent monitoring negatively affect both trust and team climate [66]. Conversely, less detailed contracts can function as a trust mechanism, fostering trust by clarifying expectations, roles, and responsibilities [67]. Woolthuis, Hillebrand, and Nooteboom observed that IPR protection and IM mutually enhance each other in project teams [68]. IPR protection mechanisms safeguard proprietary knowledge [69], thus reducing the risk and fear of exploitation. Such protection can bolster stability and predictability [69], and positively affect the team's IM [70, 71]. This primarily stems from the theory of catalytic disclosure, which posits that the sharing and disclosure of information can lead to more innovative ideas and discoveries. Nonetheless, without an appropriate IP framework, individuals might not invest their time and resources into this behavior. A strong IPP can mitigate this risk, promoting collaboration and co-creation between academics and businesses [72].

In national research projects, a robust IM culture fosters transparency in information sharing, thus increasing the likelihood of achieving RTI. RTI are enabled by employees who creatively use their knowledge and skills to integrate resources, benefiting the national research project team [73]. This integration significantly impacts the outcomes of technological innovations and the realization of radical advances [74]. Furthermore, Lusch and Nambisan contend that effective communication among team members facilitates the sharing and integration of resources and knowledge, which in turn promotes RTI [73]. Undoubtedly, a corporation with robust innovative management capabilities can encourage a proactive mindset towards exploring novel ideas and perspectives, while simultaneously providing the required resources and financial support for the implementation of technological innovations and transformation strategies. The primary reason for this is that IM can help teams overcome the challenges posed by emerging technologies such as uncertainty and complexity [75]. Furthermore, the team's visionary, adaptive, and communicative skills, which are crucial attributes influencing IM [76], serve as a solid foundation for national research project teams to foster creativity and achieve RTI.

The impact of IPP on the achievement of RTI by national research project teams is indirect, with the team's IM playing a pivotal mediating role in this process. National research project teams and members are more likely to perceive IPP as a driving force for technological innovation rather than an obstacle when working in an IM system that emphasizes high risk

tolerance, openness to new ideas, and a propensity for collaboration and learning [77]. Knowledge production within national research project teams hinges on the integration of insights from various fields and disciplines, with interorganizational teamwork serving as a vital conduit for achieving efficient and effective organizations [78, 79]. Additionally, safeguarding the workflows of interorganizational teams is crucial for fostering knowledge sharing and innovation [80]. The theory, based on the "knowledge spillover theory," asserts that IPP impede the transfer and dissemination of knowledge, leading to sub-optimal outcomes for society. This theory, however, overlooks the complexity of the IPP system and the varying degrees of exclusivity provided by IPP, as well as the intermediate factors that impact the influence of IPP on various mediating factors during innovation, such as team IM, R&D expenditures, human resource allocation, partnerships, and knowledge management practices [81, 82]. The interplay between team IM and IPP is examined. The results indicate that increased IPP levels positively impact IM within national research project teams, especially in collaborative and learning-oriented environments. Based on this, this paper proposes:

**H3a:** IPP has a significant positive effect on the IM of national research project teams.

**H3b:** The IM of national research project teams has a significant effect on RTI.

**H3c:** The IM of national research project teams has a significant mediating effect on the impact of IPP and RTI.

Stricter IPP regulations can encourage hosting national research programs to invest more in R&D. Specifically, improved patent quality enhances the chances of commercializing innovative ideas, which may lead to increased R&D expenditures [83]. Venture capitalists fund the development of promising inventions, thus advancing technological innovation and potentially impacting firms' patenting activities [84]. An econometric study by Jaffe and Trajtenberg suggests that a stronger patent system inhibits teams from increasing their R&D budget in uncertain market conditions about the future [85]. A significant association is noted between enhanced IPPs and R&DII levels among national research project teams.

Innovation acts as a vital impetus for economic growth, strategic position, and job creation. Achieving RTI for national research projects necessitates substantial financial commitments for R&D investments. Specifically, an increase in R&DII in national research project teams has been shown to positively correlate with enhanced technological performance [86]. This, in turn, leads to improved productivity and market value. Additionally, these R&D activities help to foster the acquisition of external knowledge, thereby accelerating the integration of RTI [87]. For example, Wu et al., Xu et al. studied the R&D teams of industries in China found that their R&DII has a positive impact on technological innovation [88, 89]. Sun et al. demonstrate that increased investment in R&D indirectly fosters regional green technology innovation, influenced by digital financial inclusion [90]. Alshammari, Dehghanizadeh and Faraji studied the R&D teams of industries in diverse countries' impact of financial capital investment on the realisation of RTI [91, 92].

The impact of IPPs on the R&DII of national research project teams extends beyond economic foundation building for RTIs. They also play a mediating role in IPPs and technological innovation. Good IPPs can lead to a larger scale of R&DII and foster increased innovation value through adequate resources. R&D encounters negative externalities as national research program teams often work to prevent the imitation of their intellectual property. Enhanced governmental IPP mitigates these externalities by reducing the risk of IPP infringement and boosting the expected returns on R&D investments, thereby motivating countries and firms to augment their R&D efforts [93–95]. However, this also compels firms to enhance their competitive edge and increase their innovation-driven competitiveness [96]. The primary reason is

that, based on the Appropriability Theory (AT), nations and corporations invest in R&D teams to obtain a share of the returns generated by these endeavors. This, in turn, ensures that the investment remains secure from competitors, promoting increased innovation [97]. Drawing upon the Patent Rivalry Theory (PRT), this study demonstrates that stronger IPP is crucial in reducing duplication and fostering increased R&D investments and knowledge-sharing among national research project teams [98]. For example, Liu et al. confirm that high R&DII contributes positively to RTI, especially under conditions where IPP is strengthened [99]. Based on this, this paper proposes:

**H4a:** IPP has a significant positive effect on R&DII of national research project teams.

**H4b:** R&DII of national research project teams has a significant positive effect on RTI.

**H4c:** R&DII of national research project teams has a significant mediating effect on the impact of IPP and RTI.

The success of national research projects is influenced by the team's GP. Higher GP levels are beneficial for IPP to affect RTI. Specifically, an increase in team GP positively correlates with patent generation and citations [100, 101], thus increasing the probability of RTI achievement by the national research projects team. The creativity and creative potential of researchers are crucial prerequisites for innovation. This potential is influenced not only by individual creativity but also by environmental factors [102]. Therefore, creating an environment that stimulates the innovative potential of research teams is a vital component of their innovation management. Research indicates that executives proficient in digital literacy and regions with robust IPP significantly enhance the positive effects of digital transformation on the quality of technological innovation [103].

Diverse team members, possessing varying skills, experiences, and cognitive approaches, can provide a comprehensive range of perspectives and solutions for innovation. Integrative leaders, who acknowledge and effectively integrate these diverse perspectives, can stimulate innovative possibilities. Moreover, team members demonstrating adaptability and flexibility can aid the entire team in better coping with the uncertainties inherent in the innovation process. Research indicates that GP significantly impacts team success in achieving work objectives [104]. Furthermore, GP exerts a more direct and potent influence on team empowerment and performance than individual efficacy [105]. National research teams, fueled by this GP, conduct research and innovation through effective integration of internal factors, thereby enhancing their collective research and innovation capabilities. High-potential team members are more likely to be motivated to innovate, and they exhibit heightened commitment in high-stakes R&D environments. This translated into increased effort, which in turn increases the likelihood of innovation. Moreover, these individuals can allocate resources more effectively, optimize their use, and thus enhance the efficiency of R&D, accelerating the innovation process. Conversely, the GP of a national research project team has emerged as an essential metric for assessing team performance, regardless of the potential inhibitory or promotional effects of IPP on technological innovation [88, 106]. GP in national research project teams was linked to improved team coordination, communication, and task accomplishment [107], which in turn led to higher technological innovation outcomes. This, in turn, facilitated teams to attain RTI. Based on this, this paper proposes:

**H5a:** GP acts as a moderating factor, while IL serves as a mediating factor in the impact of IPP on RTI within national research program teams.

**H5b:** GP acts as a moderating factor, while R&DII serves as a mediating factor in the impact of IPP on RTI within national research program teams.

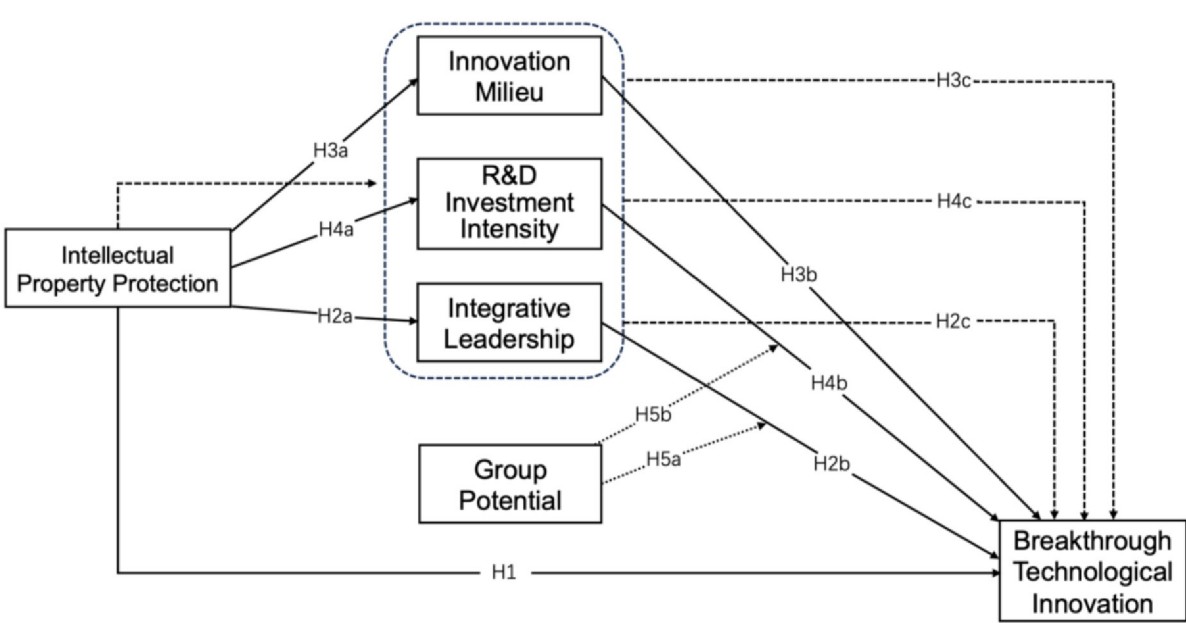

**Fig 1. Conceptual model.**

Combining hypotheses H1 to H5, the conceptual model of the study was plotted in Fig 1.

## 3. Research methods

### 3.1 Sampling and data collection

In this study, the members of the national research project teams in Chinese universities and enterprises were investigated to assess the proposed hypotheses. The data were primarily obtained from eminent universities and research organizations, such as the University of Science and Technology of China, Harbin Institute of Technology, Harbin Engineering University, Anhui University, Northeast Agricultural University, and prominent high-tech companies. Furthermore, the selection process of national research project teams involves multiple considerations, typically encompassing a diverse array of expertise from various sectors, including industry, academia, and government agencies. However, the involvement of multiple entities may lead to concerns regarding intellectual property rights, ownership, disclosure, and confidentiality. Numerous national research projects are significant financial investments funded by governments, funding bodies, academic institutions, corporations, and other stakeholders. The IPP, including patents and technological achievements, against theft and infringement is of paramount importance.

This study aims to examine the effect of IPP on the RTI of national research project teams, which is inhibited by the absence of secondary data relevant to these teams. Therefore, a questionnaire survey was conducted to gather sample data. A sample of 15 participants, consisting of 5 professors and 10 master's and doctoral students, was initially pretested to validate the questionnaire's reliability. Subsequently, the questionnaire items were slightly adjusted based on the feedback gathered from this small-sample pilot survey. This study employed three trained researchers to collect data using a random selection method. The research data was gathered from participants in national studies carried out by universities, research institutes, and corporations. A clear explanatory letter was employed to communicate the study's purpose, emphasizing the voluntary nature of participation. This paper

also guarantees to participants that their responses will remain anonymous and will only be utilized for academic purposes. The data for this study were gathered between October 2023 and November 2023. A total of 435 research questionnaires were disseminated, resulting in the receipt of 336 complete and valid questionnaires, with a validity rate of 77.2%. Table 1 presents the demographic characteristics of the study participants and the project team. It is clear that the selected sample is representative, guaranteeing the reliability, scope, and universality of the data source.

## 3.2 Measurement instruments

In the current research, seven questionnaires were utilized to examine the proposed hypotheses. The indicators within each variable's scale were chosen or modified from existing scales, in line with the study's requirements. The preliminary scale was constructed using 82 first-order observation entries. Observation variables were removed following a subsequent correlation test. From the remaining 69 observation entries, data was preserved to yield the results of subsequent scales. Descriptive statistics and correlation analyses were conducted on the variables in Table 2, which encompassed six variables: IPP, RTI, IL, IM, GP, and R&D II. The reliability and validity of the questionnaire scales in this study were evaluated through pretest and standardized measures of reliability and validity. Unlike the demographic section, all survey items were rated by respondents on a 5-point Likert scale, ranging from 1 (very low) to 5 (very high), thus encompassing various study variables. Participants provided detailed demographic information, including gender, age, educational background, and project participation duration, along with other relevant characteristics.

1. IPP. In accordance with Xiao's framework, we selected three levels of the scale, totaling 19 entries, to create an assessment scale that is consistent with the capability to IPP [108]. The

**Table 1. Demographic characteristics statistics (N = 336).**

| Variables | Frequency | Percent | Variables | Frequency | Percent |
|---|---|---|---|---|---|
| **Genders** | | | **Academic Discipline** | | |
| Male | 165 | 49.11 | Science and Engineering as Academic Subjects | 145 | 43.15 |
| Female | 171 | 50.89 | Humanities | 191 | 56.85 |
| **Age** | | | **Join the team time** | | |
| 20–30 years | 232 | 69.05 | Less than 1 year | 235 | 69.94 |
| 30–40 years | 72 | 21.43 | 1–3 years | 69 | 20.54 |
| 40–50 years | 22 | 6.55 | 3–5 years | 15 | 4.46 |
| 50–60 years | 6 | 1.79 | More than 5 years | 17 | 5.06 |
| More than 60 years | 4 | 1.19 | **Group Size** | | |
| **Educational Level** | | | Less than 10 persons | 236 | 70.24 |
| Bachelor's degree | 54 | 16.07 | 10–15 persons | 55 | 16.37 |
| Master's degree | 248 | 73.81 | 16–20 persons | 15 | 4.46 |
| Doctoral | 34 | 10.12 | More than 20 persons | 30 | 8.93 |
| **Professional Title** | | | **Project Amount ($ million)** | | |
| Junior | 43 | 12.8 | Below 30 | 229 | 68.15 |
| Intermediate | 46 | 13.69 | 30–60 | 51 | 15.18 |
| Associate | 26 | 7.74 | 60–90 | 15 | 4.46 |
| Senior | 11 | 3.27 | 90–120 | 13 | 3.87 |
| None | 210 | 62.5 | 120 or more | 28 | 8.33 |

Note: This table shows the demographic profile of the respondents to this study in relation to their project team; N = sample size.

**Table 2. Descriptive statistics and correlation analysis.** (N = 336).

| | Mean | SD | Skewness | Kurtosisi | IPP | RTI | IL | IM | GP | R&DII |
|---|---|---|---|---|---|---|---|---|---|---|
| IPP | 3.571 | 0.873 | -0.588 | 0.166 | 1 | | | | | |
| RTI | 3.248 | 0.819 | -0.366 | 0.202 | 0.666 | 1 | | | | |
| IL | 3.554 | 0.819 | -0.557 | 0.406 | 0.675 | 0.736 | 1 | | | |
| IM | 3.630 | 0.776 | -0.494 | 0.579 | 0.672 | 0.647 | 0.830 | 1 | | |
| GP | 3.653 | 0.814 | -0.463 | 0.203 | 0.633 | 0.690 | 0.805 | 0.858 | 1 | |
| R&DII | 3.387 | 0.873 | -0.300 | -0.191 | 0.679 | 0.778 | 0.696 | 0.728 | 0.791 | 1 |

Note: The table depicts the mean scores, variance, skewness and kurtosis values of the variables in the survey results, as well as correlation analyses of individual variables to test for the presence of multicollinearity problems; N = sample size.

Source: Authors' calculation.

internal consistency test for the three latent dimensions of the scale was carried out independently, resulting in Cronbach's alpha coefficients of 0.953 for IPP mechanism, 0.940 for IPP strength, and 0.941 for IPP timeliness.

2. RTI. By combining the research of Wang et al. and Wang et al., the capacity for RTI in national scientific research project teams was assessed across five dimensions: academic output, talent output, result transformation, economic output, and social benefit [109, 110]. These dimensions comprised a total of 21 entries. The internal consistency test for the five dimensions of the scale was independently conducted, resulting in Cronbach's alpha coefficients of 0.909, 0.916, 0.943, 0.918, and 0.922, respectively.

3. IL. Moreover, the approach of IL was adopted from the scale proposed by Zhang et al., three out of the ten latitudes were selected, focusing on leadership elements, strategic decision-making, and relationships, resulting in a comprehensive assessment scale for IL that is consistent with the national research project team [111]. The internal consistency tests for the three latent dimensions of the scale were independently conducted, resulting in Cronbach's alpha coefficients of 0.934, 0.899, and 0.905, respectively.

4. IM. Furthermore, the IM is described according to Liu et al.'s scale [112]. The development of an assessment scale congruent with the innovative environment of the national research project entailed selecting two dimensions of the scale consisting of six items, which were colleague support and leadership support. The internal consistency assessments for the two latitudes of the scale were carried out separately, resulting in Cronbach's alpha coefficients of 0.769 for the colleague support and 0.908 for the leadership support.

5. GP. Adopting Guzzo's scale, four dimensions were chosen: innovation culture, strategic foresight, learning ability, and sustainable benefits [113]. These formed an assessment scale to evaluate GP, in line with the national scientific research project team. Internal consistency testing was carried out, and the Cronbach's alpha coefficient was calculated to be 0.948.

6. R&DII. The R&DII Index is computed using the scale proposed by Wang et al. [109]. This scale comprises three dimensions: Infrastructure Resource Input, Human Resource Input, and Financial Resource Input. The internal consistency tests for the three latent dimensions of the scale were independently conducted, resulting in Cronbach's alpha coefficients of 0.900, 0.798, and 0.885, respectively.

### 3.3 Common method bias

To reduce common method bias (CMB) during the questionnaire completion stage, this study adopted procedural adjustments for data collection, ensuring consistency across different respondents and adhering to Podsakoff et al.'s recommendations [114]. The primary focus of the text is to provide guidelines for answering questions and fostering a deeper understanding of specific concepts. To ensure the confidentiality, anonymity, and voluntary participation of the respondents, the researcher highlighted the significance of truthful responses. Moreover, it was communicated to the participants that there were no preconceived correct or incorrect answers. Subsequently, a meticulous examination of each item was performed to verify the absence of obscure, misleading, or uncommon terms. The wording was further simplified to enhance comprehension, and the statement order was adjusted to minimize the possibility of respondent "guessing" [115]. This approach ensures the rationality and conciseness of the observations presented in this study.

Moreover, this study Utilized Harman's one-way test and Lindell & Whitney's labelled variable approach for post-hoc examination [116, 117]. Before attempting the hypothesis tests for the research model, Harman's one-way test assessed whether there was a CMB in the questionnaire data. The analysis reveals that the highest factor-explained variance was 48.089%, below the recommended 50%. However, the total explained variance of all extracted factors amounted to 73.199%, exceeding the suggested threshold level by 50%. Consequently, CMB is not an issue for this dataset, as confirmed by several recent studies [118]. Subsequently, Lindell and Whitney's approach was adopted to examine whether CMB affect the reliability of study results, utilizing theoretically uncorrelated variable as a marker (marker variable) [117]. In the ongoing study, a marker variable ("collusion with other members and general interpersonal tension") was used to assess the alignment of participants with their interpersonal styles. As indicated in Table 3, the path coefficients, p-values, and $R^2$ values of the conceptual model without marker variables show no significant differences compared to the model with marker variables, as variations are within 10% [119]. Consequently, it can be concluded that CMB did not affect the study's outcomes [117].

### 3.4 Data analysis techniques

To examine the model's assumptions, the study utilized PLS-SEM to analyze the cross-sectional data from the national research project team, and SmartPLS 4.0.9.5 software was employed to evaluate these assumptions. Structural equations and baseline regression differ in their comprehensive analysis, involving multiple independent and predictor variables. Khan et al. suggest that a structural equation model demands a minimum of 100 sample data inputs for optimal accuracy and reliability [120]. This study's valid sample data includes 336 entries, adhering to the suggested guidelines by Khan et al. [120]. Structural equation modeling offers a comprehensive analysis of the mean, variance, and covariance for each variable, simultaneously generating a composite model framework through PLS-SEM without compromising predictive accuracy. Astrachan et al. also emphasized that the application of PLS-SEM is particularly effective in managing complex model distributions and numerous indicator variables [121]. Thus, this study utilizes divergence and convergence validity measures to assess the reliability of the structural model parameters derived from SEM.

Considering the strong recommendation from the PLS literature regarding G*power analysis for determining the optimal sample size [122], the sample size estimates were analyzed using G*power 3.1.9.6 software. The sample size of our study was 336, which is notably

**Table 3. Measured marker variable test.**

| Structural Path | Baseline model | | Marker variable model | |
|---|---|---|---|---|
| | Coefficient | P-values | Coefficient | P-values |
| IPP -> RTI | 0.153 | 0.013 | 0.160 | 0.003 |
| IPP -> IL | 0.021 | 0.000 | 0.020 | 0.000 |
| IL -> RTI | 0.420 | 0.000 | 0.407 | 0.000 |
| IPP -> IL -> RTI | 0.009 | 0.000 | 0.008 | 0.001 |
| IPP -> IM | 0.029 | 0.013 | 0.021 | 0.024 |
| IM -> RTI | -0.199 | 0.008 | -0.219 | 0.001 |
| IPP -> IM -> RTI | -0.006 | 0.046 | -0.004 | 0.048 |
| IPP -> RDII | 0.018 | 0.001 | 0.019 | 0.001 |
| RDII -> RTI | 0.534 | 0.000 | 0.481 | 0.000 |
| IPP -> RDII -> RTI | 0.010 | 0.003 | 0.009 | 0.002 |
| GP x IL -> RTI | 0.121 | 0.039 | 0.110 | 0.022 |
| GP x RDII -> RTI | -0.167 | 0.005 | -0.157 | 0.003 |
| **R Square** | | | | |
| IPP | 1 | | 1 | |
| RTI | 0.718 | | 0.727 | |
| IL | 0.997 | | 0.997 | |
| IM | 0.982 | | 0.983 | |
| R&DII | 0.993 | | 0.993 | |

Source: Authors' calculation.

larger than the minimum requirement of 199 set by the G*power test. This test has an efficacy of 0.8, an alpha value of 0.05, and an effect size of 0.2. In this study, the Pearson's chi-square test for discrete variables was applied to compare early and late respondents across different variables, including age, educational background, field of study, professional title, joining time, team size, and financial investment. The research demonstrates a lack of significant difference between early and late respondents, effectively eliminating the possibility of non-response bias in this study. Additionally, this study utilizes an artificial neural network (ANN) model to enhance PLS-SEM, bypassing the latter's limitation in identifying only corrective and linear relationships [123], and ANN supplemented the non-linear relationship in the model of this study, which helps in decision making [124]. Furthermore, ANN are resilient to outliers in small samples and accommodate non-compensatory models, where a decrease in one variable does not necessitate a corresponding increase in another [125, 126]. This method enables researchers to initially apply the PLS-SEM estimation to establish linear relationships between predictors and specific target factors [127, 128]. Subsequently, based on these results, the ANN technique is employed to explore non-linear relationships [129]. The hybrid SEM-ANN approach proves effective for human behavioral studies involving complex processes and data, as demonstrated by the current study [130–132]. Based on the PLS-SEM analysis, the significant predictors were ranked considering their normalised significance [133]. The results of the ANN model were analyzed and compared with those from the PLS-SEM to assess the consistency of the predictors' relative importance derived from both methods, ensuring high accuracy. The graphical representation of the research methodology is presented in Fig 2.

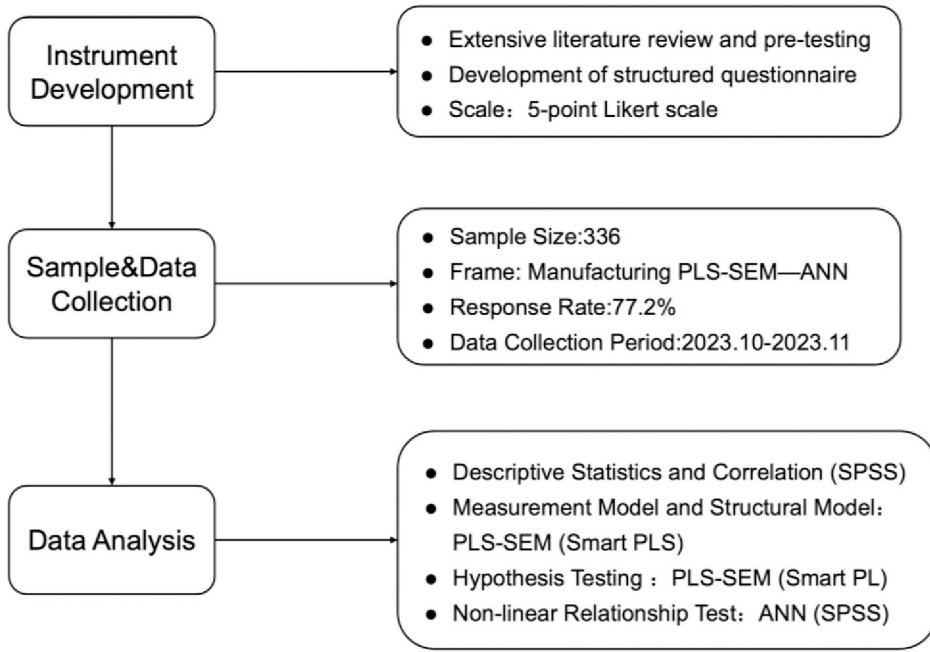

**Fig 2. Flowchart of research materials and methods.**

# 4. Results

## 4.1 Descriptive analysis

The descriptive analysis of latent structures in Table 2 reveals that the mean scores for IPP, RTI, IL, IM, GP, and R&D II are 3.571, 3.248, 3.554, 3.630, 3.653, and 3.387, respectively. According to the previous study [134], the skewness and kurtosis values of the variables are below the thresholds of ±3 and ±10, respectively. The correlation analysis demonstrated that, as depicted in Table 2, only three values among the potential structures exceeded 0.8, and their corresponding coefficients were all below 0.781. This finding suggests that the data does not suffer from severe multicollinearity issues. Additionally, based on the Variance Inflation Factor (VIF), it can be observed that the discriminant validity (DV) criterion is met, indicating that the model is suitable for further statistical analysis.

## 4.2 Measurement model

This research Utilized four assessments to construct convergent validity, discriminant validity, item-level reliability, and internal consistency reliability. As illustrated in Table 4, the minimal factor loading was 0.625, and the maximal factor reached 0.902. These numbers exceed the suggested threshold of 0.50 [135]. The study indicates the robustness of individual items in this research. If the external loading value lies between 0.40 and 0.50, the researcher can utilize the item without it influencing the retrieved composite reliability (CR) and average variance (AVE) significantly. As per Nunnally and Bernstein's recommendation, the Cronbach's alpha and CR of each variable are utilized to determine internal consistency reliability [136]. It is indicated by Nunnally and Bernstein that a Cronbach's alpha value should be above the threshold of 0.7 [136]. In this study, the Cronbach's alpha value ranges from 0.769 to 0.974, exceeding the threshold of 0.7, suggesting strong internal consistency [137]. According to Hair et al., a

**Table 4. Construct validity and reliability.** (*N* = 336).

| | Items | Factor Loadings | Alpha | CR | AVE |
|---|---|---|---|---|---|
| **IPP** | | | 0.974 | 0.976 | 0.681 |
| **In terms of mechanisms for the protection of intellectual property rights, the actual situation of your team is that** | IPP1 | | 0.953 | 0.960 | 0.726 |
| Extent of establishment and improvement of patent early warning mechanisms | IPP11 | 0.784 | | | |
| Familiarity with patent protection routes and procedures | IPP12 | 0.796 | | | |
| Familiarity with the means and procedures for the protection of exclusive trademark rights | IPP13 | 0.784 | | | |
| Proportion of confidentiality agreements with classified employees | IPP14 | 0.752 | | | |
| Degree of sophistication of confidentiality mechanisms for study tours for outsiders | IPP15 | 0.840 | | | |
| Degree of sophistication of confidentiality mechanisms in negotiations with clients | IPP16 | 0.828 | | | |
| Degree of sophistication of confidentiality mechanisms in collaborative R&D or manufacturing | IPP17 | 0.814 | | | |
| Degree of sophistication of the review mechanism for works published, e.g., through journals or conferences | IPP18 | 0.807 | | | |
| Familiarity with copyright protection routes and procedures | IPP19 | 0.824 | | | |
| **In terms of the strength of intellectual property protection, the reality for your team is that** | IPP2 | | 0.940 | 0.954 | 0.807 |
| Degree of combined protection of core and peripheral patents | IPP21 | 0.842 | | | |
| Strength of legal liability for infringement of patent rights by others | IPP22 | 0.860 | | | |
| Ability to prevent confusion or dilution by combining primary and secondary marks | IPP23 | 0.828 | | | |
| Strength of legal liability for infringement of the exclusive right to use a trademark by others | IPP24 | 0.851 | | | |
| Strength of legal liability for copyright infringement | IPP25 | 0.866 | | | |
| **In terms of the statute of limitations for the protection of intellectual property rights, the reality for your team is that** | IPP3 | | 0.941 | 0.955 | 0.809 |
| Timeliness and effectiveness of the current legal system to protect patents | IPP31 | 0.839 | | | |
| Validity of Domain Name Protection in Connection with Registered Trademarks or Trade Names | IPP32 | 0.846 | | | |
| Timeliness in responding to infringements of exclusive trademark rights by others | IPP33 | 0.82 | | | |
| Strength of legal accountability for infringement of technical secrets | IPP34 | 0.839 | | | |
| Timeliness and effectiveness of the current legal regime for copyright protection | IPP35 | 0.848 | | | |
| **RTI** | | | 0.973 | 0.975 | 0.653 |
| **In terms of the academic outputs of the NRP team, the reality for your team is that** | RTI1 | | 0.909 | 0.932 | 0.734 |
| Extent of the number of patent outputs of the team | RTI11 | 0.737 | | | |
| Extent of the team's output of papers and monographs | RTI12 | 0.685 | | | |
| Ratio of highly cited papers for team papers | RTI13 | 0.793 | | | |
| Ratio of Chinese high-level papers to team papers | RTI14 | 0.758 | | | |
| Ratio of high-level foreign-language papers to team papers | RTI15 | 0.756 | | | |
| **In terms of the output of talent from national research project teams, the reality of your team is that** | RTI2 | | 0.916 | 0.941 | 0.798 |
| Percentage of national-level senior personnel produced by the team | RTI21 | 0.792 | | | |
| Percentage of teams that have produced provincial and ministerial-level senior talents | RTI22 | 0.798 | | | |
| Percentage of teams producing high-level talent | RTI23 | 0.827 | | | |
| Percentage of teams producing industry-competitive talent | RTI24 | 0.791 | | | |
| **In terms of translating the results of national research project teams, the actual situation of your team is that** | RTI3 | | 0.943 | 0.954 | 0.777 |
| Degree of industrialisation of the project team's innovations | RTI31 | 0.842 | | | |
| Ability to translate the team's innovations | RTI32 | 0.819 | | | |
| Speed of development of technological innovations for projects | RTI33 | 0.828 | | | |
| Amount of patents sold by the team | RTI34 | 0.83 | | | |
| Actual contract amount for team technology transfer | RTI35 | 0.833 | | | |
| Science and technology funding entrusted by enterprises and institutions | RTI36 | 0.824 | | | |
| **In terms of the economic output of the national research project team, the actual situation of your team is that** | RTI4 | | 0.918 | 0.948 | 0.859 |
| Degree of production of project output per team member | RTI41 | 0.864 | | | |
| Ratio of technology income from team innovations | RTI42 | 0.838 | | | |

*(Continued)*

**Table 4.** (Continued)

| | Items | Factor Loadings | Alpha | CR | AVE |
|---|---|---|---|---|---|
| Technology market turnover ratio per team member | RTI43 | 0.839 | | | |
| **In terms of the social benefits of national research project teams, the reality of your team is that** | RTI5 | | 0.922 | 0.951 | 0.866 |
| Degree of output of major basic research results of the team | RTI51 | 0.837 | | | |
| Number of outputs of significant practical outcomes for the team | RTI52 | 0.832 | | | |
| Number of awards for national scientific research achievements of the team | RTI53 | 0.829 | | | |
| **IL** | | | 0.963 | 0.968 | 0.730 |
| **In terms of integrating the leadership elements of a national research project team, the reality for your team is that** | IL1 | | 0.934 | 0.953 | 0.834 |
| Level of inspiration and motivation for team members | IL11 | 0.849 | | | |
| Strength of coordination and guidance of the team | IL12 | 0.865 | | | |
| Alignment of the innovation team's behaviours with the vision and mission | IL13 | 0.862 | | | |
| Leadership skills and technical competencies possessed | IL14 | 0.870 | | | |
| **As far as the integration of strategic decisions of national research project teams is concerned, the actual situation of your team is that** | IL2 | | 0.899 | 0.937 | 0.833 |
| Alignment of goals between leaders and members | IL21 | 0.886 | | | |
| The extent to which leaders and members of innovation teams make and plan decisions together | IL22 | 0.839 | | | |
| Leaders' capacity for strategy development | IL23 | 0.853 | | | |
| **As far as the integration of relationships in national research project teams is concerned, the actual situation of your team is that** | IL3 | | 0.905 | 0.941 | 0.841 |
| Ability to deal with the relationship between research projects and stakeholders | IL31 | 0.848 | | | |
| Partnership-building capacity for innovative teams | IL32 | 0.840 | | | |
| Sustainability of partnerships for innovative teams | IL33 | 0.853 | | | |
| **IM** | | | 0.930 | 0.943 | 0.676 |
| **In terms of colleague support for national research project teams, the reality for your team is that** | IM1 | | 0.769 | 0.866 | 0.685 |
| Team members often share different views and opinions with each other | IM11 | 0.771 | | | |
| Team members are not wary of others stealing their skills and abilities from each other | IM12 | 0.625 | | | |
| Team members are generally able to communicate and coordinate with each other to resolve problems and conflicts | IM13 | 0.876 | | | |
| **In terms of leadership support for national research project teams, the reality for your team is that** | IM2 | | 0.908 | 0.942 | 0.844 |
| Leaders are able to respect and tolerate different opinions and disagreements from employees | IM21 | 0.853 | | | |
| Leaders encourage their subordinates to innovate and share their failures | IM22 | 0.841 | | | |
| Leaders are able to trust their subordinates and delegate appropriately | IM23 | 0.858 | | | |
| **GP** | | | 0.948 | 0.957 | 0.762 |
| **In terms of the group potential of the members of a national research project team, the reality of your project is that** | GP1 | | 0.948 | 0.957 | 0.762 |
| I have confidence in the team's overall scientific ability | GP11 | 0.867 | | | |
| I think the team has strong project organisation skills | GP12 | 0.865 | | | |
| I think the team has a strong level of innovation in scientific tasks | GP13 | 0.879 | | | |
| I feel that the team was able to solve the problems and issues encountered | GP14 | 0.869 | | | |
| I feel that the team has a strong ability to learn and research at the forefront of their discipline | GP15 | 0.881 | | | |
| I feel that the team was able to achieve a high level of scientific results | GP16 | 0.879 | | | |
| The overall capacity of our research team is sufficient to cope with the scientific tasks on many important topics. | GP17 | 0.871 | | | |
| **R&DII** | | | 0.944 | 0.954 | 0.748 |
| **In terms of the investment of infrastructure resources in national research project teams, the actual situation in your team is that** | R&DII1 | | 0.900 | 0.953 | 0.909 |
| Ratio of inputs from research and development organisations in national research projects | R&DII11 | 0.902 | | | |
| Input ratio of government-industry-university-research co-operation bases in national scientific research projects | R&DII12 | 0.896 | | | |
| **In terms of human resource inputs in national research project teams, the actual situation in your team is that** | R&DII2 | | 0.798 | 0.908 | 0.832 |

(Continued)

**Table 4.** (Continued)

| | Items | Factor Loadings | Alpha | CR | AVE |
|---|---|---|---|---|---|
| Ratio of inputs of scientifically and technologically active persons in innovation teams | R&DII21 | 0.857 | | | |
| Input ratio of the number of people with middle and senior titles in the innovation team | R&DII22 | 0.797 | | | |
| **In terms of financial resources invested in national research programmes, the actual situation of your team is that** | R&DII3 | | 0.885 | 0.946 | 0.897 |
| Extent of government funding for innovation in national research programmes | R&DII31 | 0.870 | | | |
| Extent of internal expenditure of R&D funds in national research projects | R&DII32 | 0.859 | | | |

Note: The table shows the reliability of the individual items of the questionnaire scale and the accuracy and validity of the model structure as judged by four tests;

Alpha = Cronbach's Alpha, CR = Composite reliability, AVE = Average variance extracted; N = sample size.

Source: Authors' calculation.

CR value of more than 0.6 is mandatory [122]. In the exploratory survey, a CR value between 0.60 and 0.70 is considered acceptable. Values between 0.70 and 0.95 are regarded as good, and those exceeding 0.95 are considered the best. The results in Table 2 reveal that all structures have CR values higher than 0.866, suggesting a model structure with good accuracy and reliability, which meets the internal consistency test [138]. Convergent validity pertains to the measure of how variable indicators mirror the same underlying structure. The data in Table 4 demonstrates that the minimal AVE value stands at 0.653, whereas the maximum value reaches 0.909. Therefore, this study adheres to the convergent validity standard set by Hair et al., which necessitates an AVE value of no less than 0.50 [122].

DV is a situation where researchers find it difficult to differentiate between two indicators due to their lack of statistical uniqueness [139]. Fornell and Larcker introduced two standard indicators for calculating DV through two separate methods [140]. The first approach involves comparing the square root of AVE with the correlation statistic, while the second approach entails comparing the AVE value with the squared correlation value. In recent years, researchers have devised a novel technique for calculating DV, which shows the shortcomings of the previous metrics. Henseler et al. introduced the heterotrait-monotrait ratio (HTMT) correlation as a novel method for calculating DV [141]. This study also utilizes the Fornell-Larcker criterion and HTMT criterion. Studies by others suggest that the square root of the AVE for each structure should be greater than the correlation coefficient for each row, as shown in Table 5, to validate the discriminant validity of the structure. According to Henseler et al. it

**Table 5. Fornell–Larcker criterion.**

| | IPP | RTI | IL | IM | GP | R&DII |
|---|---|---|---|---|---|---|
| IPP | **0.825** | | | | | |
| RTI | 0.666 | **0.808** | | | | |
| IL | 0.677 | 0.739 | **0.855** | | | |
| IM | 0.676 | 0.643 | 0.82 | **0.822** | | |
| GP | 0.636 | 0.693 | 0.804 | 0.859 | **0.873** | |
| R&DII | 0.683 | 0.780 | 0.696 | 0.727 | 0.793 | **0.865** |

Note: The table shows the results of comparing the square root values of AVE for each variable with their correlation coefficients for each row; Bold values on the correlation matrix's diagonal are AVE's square roots. Off-diagonal elements below the diagonal are correlations among the constructs.

Source: Authors' calculation.

**Table 6. HTMT criterion.**

|  | IPP | RTI | IL | IM | GP | VIF |
|---|---|---|---|---|---|---|
| RTI | 0.684 |  |  |  |  | 2.639 |
| IL | 0.698 | 0.761 |  |  |  | 1.922 |
| IM | 0.707 | 0.679 | 0.877 |  |  | 1.867 |
| GP | 0.660 | 0.719 | 0.842 | 0.915 |  | 1.066 |
| R&DII | 0.709 | 0.812 | 0.731 | 0.777 | 0.837 | 1.887 |

Note: The table shows the HTMT values versus VIF values for each structure to further determine multicollinearity; VIF = Variance inflation factor.
Source: Authors' calculation.

was shown that the theoretical threshold of HTMT is 0.9 for theoretically identical constructs and 0.85 for conceptually different variables [141]. As depicted in Table 6, all constructions have an HTMT of less than 0.9, except for the HTMT between IM and GP, which is 9.15. This could imply a minor covariance problem between the innovation environment and the group's potential relationship. Nevertheless, according to Hair et al. demonstrated is the use of the VIF for assessing multicollinearity, with its value being below 5 being crucial [122]. The analysis showcases that all VIF values in Table 6 fall beneath this limit, fulfilling the DV criterion. Therefore, the study's multicollinearity issue is resolved, paving the way for further analyses.

Table 7 illustrates the predictive relevance of the structure, reflecting the predictive potential of the model's predictor variables. In structural equation modeling, the model's predictive power is evaluated based on two criteria: predictive accuracy and predictive relevance. Hair et al. (2014) and Khan, Hashim, and Bhutto clarified that predictive accuracy assesses the explained variance of each endogenous construct using the coefficient of determination ($R^2$) [142, 143]. $R^2$ and $Q^2$, as indicators of predictive power, are employed respectively according to Cohen; a value of $R^2$ exceeding 0.26 is considered valuable [144]. The $R^2$ values for the variables IPP, RTI, IL, IM, and RD&II in Table 6 are 1, 0.718, 0.997, 0.982, and 0.993, respectively, indicating the variables' strong predictive abilities [145–147]. Additionally, $Q^2$'s magnitude signifies the importance of the endogenous component. This study was evaluated using the $Q^2$ value, calculated through the blindfolding procedure in SmartPLS 4.0 software, with a value greater than 0 indicating its relevance in prediction [141]. The study reveals that the Q2 values for the predictive relevance indicators of IPP, RTI, IL, IM, and RD&II were 0.603, 0.577, 0.625, 0.625, and 0.661, respectively. This indicates that the model's predictive relevance is adequate

**Table 7. Predictive relevance of the model.**

|  | R Square | $Q^2$( = 1-SSE/SSO) |
|---|---|---|
| Intellectual Property Protection | 1 | 0.603 |
| Radical Technology Innovation | 0.718 | 0.577 |
| Integrated Leadership | 0.997 | 0.625 |
| Innovation Milieu | 0.982 | 0.625 |
| R&D Investment Intensity | 0.993 | 0.661 |

Note: The table shows the predictive relevance of each structure, and according to the two predictive ability indicators, R2 and Q2, it shows that the model in this study has very good predictive ability; R Square = $R^2$, $Q^2$ = Q Square.
Source: Authors' calculation.

[146]. Additionally, the SRMR values from the PLS-SEM were utilized to evaluate the model's fit, and the SRMR coefficient for the model in this research was 0.053, which fell short of the 0.1 threshold. Consequently, the model exhibits a satisfactory fit [148].

## 4.3 Structural model

After validating the measurement model, the study employed SmartPLS 4.0.9.5 to test the hypotheses associated with the research model. By calculating p-values and t-values, the significance of the proposed hypotheses was assessed. If the t-value surpasses 1.96 or the p-value falls below 0.05, the hypothesis is accepted, otherwise it is rejected. In the current research, we utilized PLS-SEM in conjunction with the bootstrapping method available in SmartPLS to select a bootstrap sample consisting of 5000 observations from the raw data. The purpose of this selection process was to evaluate the significance of the path coefficients, and the findings of the hypothesis testing are depicted in Fig 3 and Table 8.

The research demonstrates the importance of the 12 proposed relationships within the model. As depicted in Table 8, IPP has a positive impact on national research project team RTI (β = 0.153, t = 2.488, p = 0.013), which supports the hypothesis H1. The path coefficients in Table 8 indicate that a 1% increase in IPP leads to a 0.153% increase in RTI. This study moreover acknowledged that IPP has a considerable positive effect on IL (β = 0.021, t = 3.756, p = 0.000). The path coefficients indicate that a 1% rise in IPP results in a 0.021% increase in IL, thus reinforcing hypothesis H2a. The research demonstrates a substantial positive influence of IL on the research and development impact RTI of national research project teams (β =

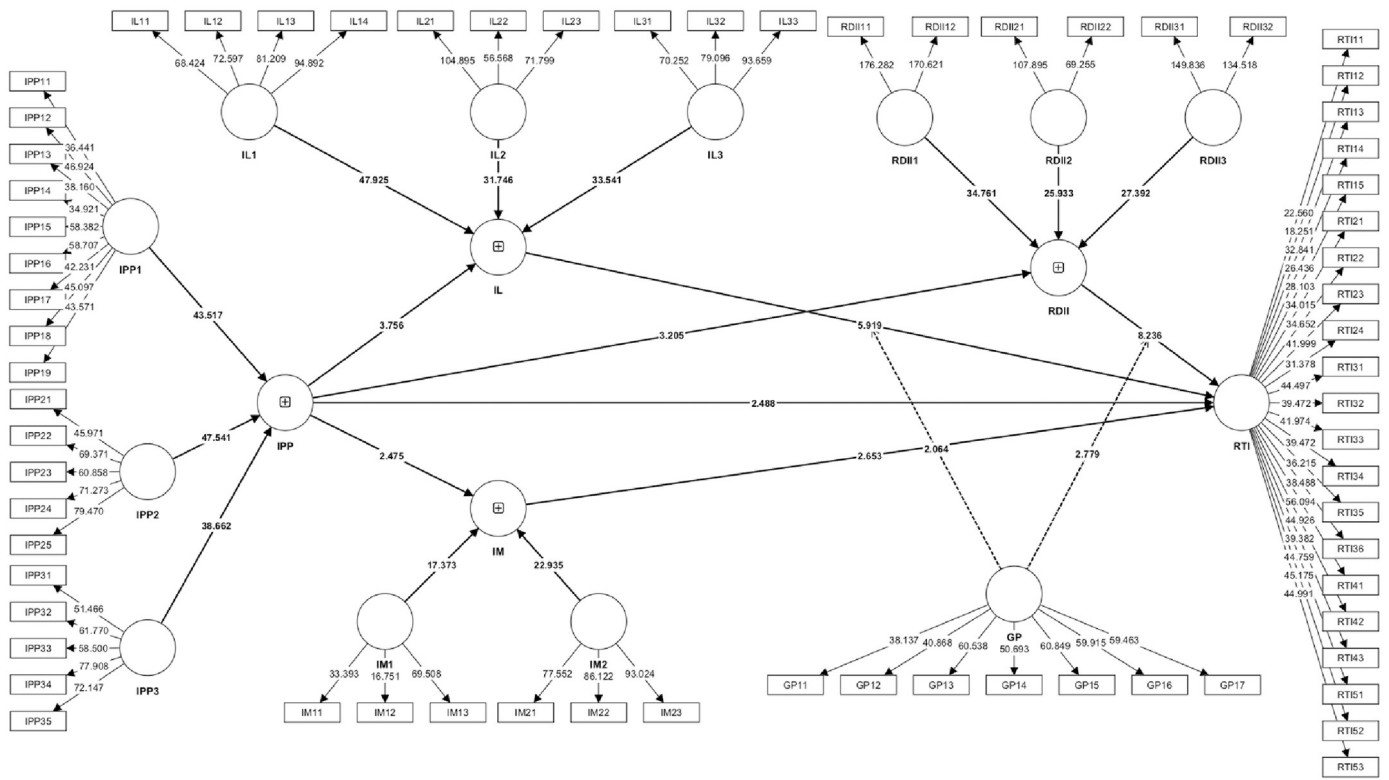

**Fig 3. Structural model.**

Table 8. Results of hypothesis testing.

| Hypothesis | Structural Path | Coefficient | t-statistics | P-values | Remarks |
|---|---|---|---|---|---|
| H1 | IPP -> RTI | 0.153 | 2.488 | 0.013 | Supported |
| H2a | IPP -> IL | 0.021 | 3.756 | 0.000 | Supported |
| H2b | IL -> RTI | 0.420 | 5.919 | 0.000 | Supported |
| H2c | IPP -> IL -> RTI | 0.009 | 3.669 | 0.000 | Supported |
| H3a | IPP -> IM | 0.029 | 2.475 | 0.013 | Supported |
| H3b | IM -> RTI | -0.199 | 2.653 | 0.008 | Supported |
| H3c | IPP -> IM -> RTI | -0.006 | 1.992 | 0.046 | Supported |
| H4a | IPP -> R&DII | 0.018 | 3.205 | 0.001 | Supported |
| H4b | R&DII -> RTI | 0.534 | 8.236 | 0.000 | Supported |
| H4c | IPP -> R&DII -> RTI | 0.010 | 2.953 | 0.003 | Supported |
| H5a | GP x IL -> RTI | 0.121 | 2.064 | 0.039 | Supported |
| H5b | GP x R&DII -> RTI | -0.167 | 2.779 | 0.005 | Supported |

Note: The table shows the results of each hypothetical path test, including the direct path and each of the mediating and moderating paths.

Source: Authors' calculation.

0.420, t = 5.919, p = 0.000). The coefficients indicate that a 1% increase in IL results in a 0.42% increase in RTI, thereby supporting hypothesis H2b. While IPP holds significant influence within the national research project team, IL serves as a vital mediator ($\beta$ = 0.009, t = 3.669, p = 0.000), supporting hypothesis H2c.

Moreover, IPP demonstrates a sturdy positive effect on IM ($\beta$ = 0.029, t = 2.475, p = 0.013), validating hypothesis H3a. The findings indicate a strong correlation between IM and the RTI of the national research project team ($\beta$ = -0.199, t = 2.653, p = 0.008). The coefficients imply that an increase in the RTI of the national research project team corresponds to a corresponding increase in IM (by 0.199%), thereby supporting H3b. Furthermore, IM displays a mediating effect on the connection between IPP and RTI within the national research project team ($\beta$ = -0.006, t = 1.992, p = 0.046). The coefficients indicate that the inclusion of the mediating variable IM enhances the impact of IPP on RTI. This suggests that when reassessing the relationship between IPP and RTI, RTI also has some influence on IPP, albeit to a lesser extent, thus confirming H3c.

Moreover, IPP displays a modest yet substantial positive impact on R&DII ($\beta$ = 0.018, t = 3.205, p = 0.001), thus strengthening H4a. In contrast, R&DII demonstrates a significant positive effect on the RTI of national research project teams ($\beta$ = 0.534, t = 8.236, p = 0.000). Among the hypothesized relationships, the most significant factor directly affecting RTI is the R&DII, which exhibits a 0.534 percent increase for every 1 percent increase in R&DII, supporting H4b. In the case that IPP plays an influencing role on national research project teams, R&DII also plays a significant mediating role ($\beta$ = 0.010, t = 2.953, p = 0.003), therefore, hypothesis H4c is supported.

The moderating effect of GP on the impact of IL in IPP on the RTI of national research project teams was investigated by incorporating additional moderating variables into the mediating effect ($\beta$ = 0.121, t = 2.064, p = 0.039). Upon introducing the moderating effect of GP, the path coefficient of the mediating effect of IL in the impact of IPP on RTI increased from 0.009 to 0.121, thus, the moderating variable GP significantly enhanced the effect on RTI of national research project teams, supporting H5a. GP also showed a moderating effect for R&DII in the mediation of IPP on RTI for national research project teams, with the path coefficient of -0.167, a t-value of 2.779, and a p-value of 0.005. This supports H5b.

## 4.4 ANN analysis

(1) Analysis of neural network outputs. The term "ANN" refers to a "massively parallel distributed processor" composed of "simple processing units with neural tendencies to store and utilize experimental knowledge" [149]. Empirical findings indicate that ANN outperforms conventional regression techniques [150]. The PLS-SEM model does not address nonlinear relationships [151]. Consequently, an ANN analysis was employed both to validate the PLS-SEM results and to explore nonlinear associations. The integration of PLS-SEM with ANN aims to demonstrate that relationships are not only linear but also compensatory [152]. Additionally, ANN deliver highly accurate results by detecting linear and nonlinear correlations among variables [151]. ANN acquire knowledge through iterative training, analogous to the human brain, and encode this knowledge in synaptic weights [153]. An activation function regulates these weights, minimizing the discrepancy between the actual and desired outputs through iterative adjustments. Researchers have applied ANN techniques across various fields, including accounting [154, 155], management [151], and tourism [156]. In the current research, ANN analysis was performed with the aid of SPSS26 software, and variables significant according to PLS-SEM were included in the ANN analysis. Consequently, the IPP, RTI, IL, IM, R&DII, and GP variables were considered relevant. Fig 4 displays the ANN model incorporating one output neuron (RTI) and five input neurons (IPP, IL, IM, GP, and R&DII).

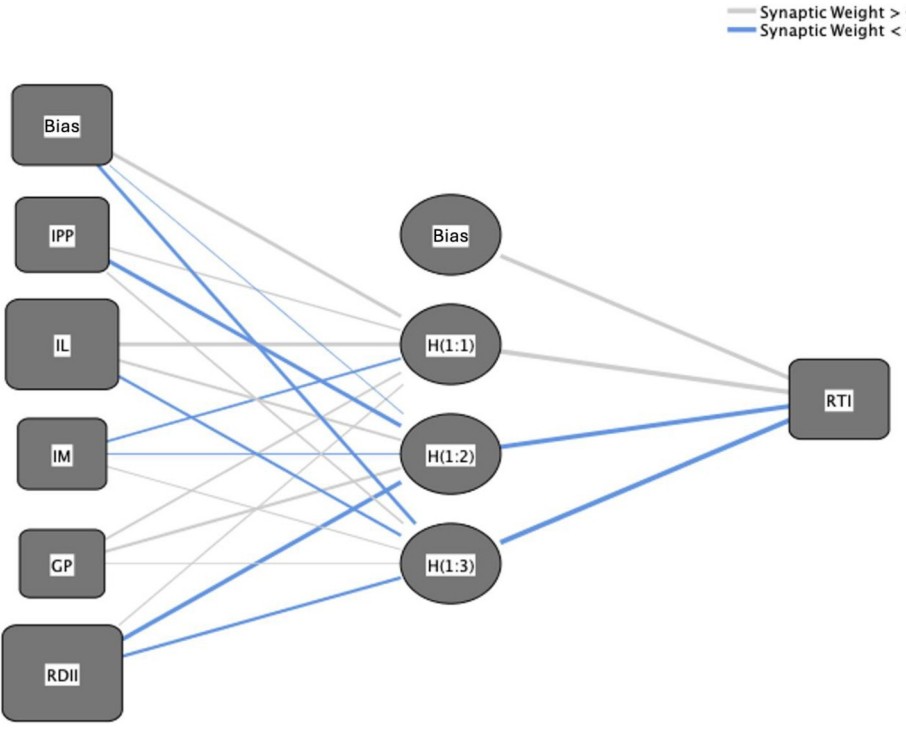

Hideen layer activation function: Sigmoid

Output layer activation function: Sigmoid

**Fig 4. Artificial neural network diagram.**

In this study, a deep ANN with a hidden layer for each output neuron node was utilized [124]. The number of neurons in the hidden layer is autonomously generated by the SPSS neural network module [153, 157, 158]. To improve the prediction accuracy, a sigmoid function was applied to stimulate both output and hidden neurons. The established interval between input and output neurons was set to [0,1] [159]. To reduce overfitting in the ANN model, this study employed a 90:10 ten-fold cross-validation technique to train and validate the collected data [160]. The data were divided into two segments: 90% for training the model and 10% for testing it [161, 162]. The predictive accuracy of the ANN model was assessed using the root mean square error (RMSE) values from both the training and test datasets [153, 163]. Hyndman and Koehler note that the RMSE value is always positive, with a value of zero indicating perfect accuracy [163]. The model's training and testing phases, utilizing the derived data, yielded mean RMSE values of 0.08137 and 0.07596, respectively (as presented in Table 9). The RMSE metrics are utilized to assess the neural network model's accuracy [164]. The RMSE is calculated using the formula: $RMSE = \sqrt{\frac{SSE}{N}}$, where SSE denotes the sum-of-square error of the training or test data, and N represents the number of samples for the training or test data. The RMSE values for the test and training data in the ANN model analysis were relatively small, indicating a high predictive accuracy for the ANN model in this study [165]. As suggested by Leong et al., ANN models have excellent predictive accuracy and matching [162]. The RMSE values of the model developed in this study are lower than those reported by predictive models developed in previous studies such as Ibrahim et al. and Alkawsi et al. [146, 161].

(2) Sensitivity analysis. Subsequently, this study performed a sensitivity analysis to rank the exogenous components according to their normalized relative importance in relation to the endogenous structure, as depicted in Table 10. To determine the normalized normalized importance of each neuron in this research, the relative importance of each neuron was calculated by dividing it by its maximum importance, and this was subsequently reported as a percentage outcome. Among the variables, R&DII displays the greatest predictive capability for the RTI of the NRP team, followed by IL, IM, IPP, and GP. The normalized importance values

**Table 9. RMSE values.**

| Network | Training | | | Testing | | | Total Samples |
|---|---|---|---|---|---|---|---|
| | N | SSE | RMSE | N | SSE | RMSE | |
| 1 | 298 | 1.942 | 0.08073 | 38 | 0.181 | 0.06902 | 336 |
| 2 | 306 | 2.046 | 0.08177 | 30 | 0.158 | 0.07257 | 336 |
| 3 | 304 | 1.993 | 0.08097 | 32 | 0.204 | 0.07984 | 336 |
| 4 | 298 | 2.171 | 0.08535 | 38 | 0.122 | 0.05666 | 336 |
| 5 | 293 | 1.916 | 0.08087 | 43 | 0.187 | 0.06595 | 336 |
| 6 | 305 | 2.072 | 0.08242 | 31 | 0.170 | 0.07405 | 336 |
| 7 | 303 | 2.077 | 0.08279 | 33 | 0.214 | 0.08053 | 336 |
| 8 | 296 | 1.912 | 0.08037 | 40 | 0.245 | 0.07826 | 336 |
| 9 | 301 | 1.705 | 0.07526 | 35 | 0.401 | 0.10704 | 336 |
| 10 | 303 | 2.094 | 0.08313 | 33 | 0.189 | 0.07568 | 336 |
| Mean | | 1.993 | 0.08137 | | 0.207 | 0.07596 | |
| SD | | 0.132 | 0.003 | | 0.076 | 0.013 | |

Note: The table shows the data results of running the training and testing of the ANN model using the ten-fold cross-validation approach; SSE = Sum square of errors, RMSE = Root mean square of errors, and N = sample size.
Source: Authors' calculation.

**Table 10.  Sensitivity analysis.**

| Neural Network(NN) | IPP | IL | IM | GP | R&DII |
|---|---|---|---|---|---|
| NN(i) | 0.107 | 0.316 | 0.146 | 0.086 | 0.346 |
| NN(ii) | 0.071 | 0.379 | 0.122 | 0.038 | 0.389 |
| NN(iii) | 0.077 | 0.386 | 0.103 | 0.046 | 0.388 |
| NN(iv) | 0.104 | 0.251 | 0.145 | 0.156 | 0.344 |
| NN(v) | 0.133 | 0.329 | 0.086 | 0.05 | 0.403 |
| NN(vi) | 0.034 | 0.399 | 0.046 | 0.025 | 0.496 |
| NN(vii) | 0.034 | 0.361 | 0.139 | 0.006 | 0.461 |
| NN(viii) | 0.118 | 0.336 | 0.107 | 0.057 | 0.382 |
| NN(ix) | 0.059 | 0.374 | 0.122 | 0.048 | 0.398 |
| NN(x) | 0.159 | 0.26 | 0.5 | 0.4 | 0.49 |
| Average importance | 0.0896 | 0.3391 | 0.1516 | 0.0912 | 0.4097 |
| Normalized importance(%) | 22.46 | 83.64 | 27.21 | 14.54 | 100 |

Note: The table shows the results of the sensitivity analysis of the ANN model, yielding an order of importance for the model predictors.

Source: Authors' calculation.

are 100%, 83.64%, 27.21%, 22.46%, and 14.54% for R&DII, IL, IM, IPP, and GP, respectively. The results from the ANN model align with those derived from the PLS-SEM model.

## 5. Discussion

This study draws upon innovative-driven and AMO theories to empirically examine the role of IPP in promoting RTI within national research project teams. It also investigates the mediating effects of IL, IM, and R&DII in this context, while incorporating GP to regulate the roles of IL and R&DII in improving RTI at the national level.

The results of this study confirm hypothesis H1. The PLS-SEM model and ANN method analysis suggest that IPP enhances the RTI in national research projects. Moreover, academic research has shown that the IPP through the judiciary has a considerable positive impact on enterprise digital innovation and plays a more significant role in promoting underlying digital technological advancements. Particularly, small businesses are more vulnerable to the influence of IPP on their digital innovation compared to large enterprises [166]. Zheng et al. focus their analysis solely on the impact of external institutional factors on technological innovation, considering the internal dynamics of firms and research teams [166]. In contrast, this study broadens the scope by introducing a new model that comprehensively incorporates internal team factors, potentially influencing RTI.

The hypothesis H2a suggests that IPP has a substantial positive effect on IL within national research project teams, a finding confirmed by past studies. IL may be influenced by the IP system during decision-making, thus they need to take IPP into account when developing strategies to utilise the organisation's innovative capacity [167]. The research endorses Hypothesis H2b by emphasizing the crucial role of relational integration and organizational cooperation in IL, in utilizing complementary resources for innovation, ultimately leading to technological innovation [166]. This innovative study expands the understanding of the impact of IPP on the RTI of national research project teams by introducing the mediating role of IL. The hybrid SEM-ANN results reveal IL as a crucial predictor of RTI achievement. Furthermore, recent studies have emphasized the role of IL in enhancing business potential and innovation outcomes, as well as improving Small and Medium-sized Enterprises' (SMEs') green technological

innovation capabilities [168]. This study diminishes the direct role of IL in RTI by treating it solely as a mediating variable. However, it expands the understanding of the factors influencing RTI by situating it within a complex framework.

In addition, IPP favours national research project team IM (H3a), a result that is consistent with previous research findings that better IPP can build positive IM [7]. The IM-led national research project team has a significant impact on RTI (H3b), a finding also supported by previous studies. These studies suggest that IM significantly boosts firm innovation capabilities and offers a partial solution to the shortcomings of formal systems [169]. In the end, IM serves as a mediator in the relationship between IPP and RTI (H3c). Numerous studies have demonstrated that IM enhances IPP and boosts RTI. More recent research suggests that IPP could promote more open and collaborative IM, thereby increasing the effectiveness of RTI [170].

As proposed in H4a, IPPs have a positive effect on national research project teams R&DII, a finding in line with previous mainstream literature [171]. Recent studies indicate that weaker national IPPs might limit R&D investments by certain firms and potentially distort technology availability within a country [172]. The results from the PLS-SEM analysis align with those from the ANN analysis, demonstrating that the R&DII is the most significant predictor of RTI in national research project (H4b). In line with previous studies that emphasize the crucial role of R&DII in promoting RTI [7, 173]. The findings confirm that the linear relationships identified by the PLS-SEM are corroborated by the nonlinear relationships discerned through the ANN model. These results provide more substantive insights than previous studies, which solely focused on linear relationships between structures [145, 174–176]. Consequently, this research underscores the theoretical implications of employing the SEM-ANN predictive analytics framework, a novel methodological paradigm within transportation engineering literature. Moreover, this study reveals a nonlinear relationship between the predictor and target structures, reinforcing the validity of the linear analyses conducted using PLS-SEM. R&DII functions as a mediator in the association between IPP and RTI (H4c), as per Song and Chen [7]. They emphasize that the relationship between R&D investment, which is influenced by IPP, displays non-linearity in relation to green innovation capacity within manufacturing.

This study emphasizes the capability of the national research project team's GP to fine-tune the mediating function of IL in the impact of IPP on RTI (H5a) and the potential to alter the mediating role of R&DII in the effect of IPP on RTI (H5b). This argument is reinforced by earlier research on GP's involvement in RTI, including studies by Xu et al., among others [158]. In this study, the analysis of both the PLS-SEM and the ANN consistently indicates that the key predictors of RTI in national research project teams are ranked in descending order of importance as follows: R&DII, IL, IM, IPP, and GP. This study supports the conclusions drawn from diverse economic and industrial contexts of previous studies. The authors claim that this is the first research to explicitly demonstrate the impact of IPPs on the RTI of national research project teams, mediated by IL, IM, and RD&II, with GP serving as a moderator. Regarding the implications, the findings hold significant relevance for policymakers and researchers involved in national R&D projects, emphasizing the crucial role of IPPs in fostering innovation and enhancing the overall effectiveness of these projects. Additionally, the study highlights the need for further exploration of the relationship between IPPs, RTI, and national research project teams, especially in the context of emerging economies. The factors influencing RTI in national research project teams extend beyond the immediate project parameters to include economic, trade, and market dimensions. Currently, these teams are predominantly based in universities and research institutes, with only a minor presence in enterprises. This represents a gap that future research could address by enhancing the focus on corporate involvement.

## 6. Conclusions and implications

The primary objective of this study is to investigate the enhancing effect of IPP on the RTI of national research project teams. The results from the analyses conducted using the PLS-SEM model and the ANN method suggest that the "moderate enhancement" of IPP can encourage national research project teams to actively explore innovations and protect their intellectual assets. This, in turn, enhances their competitive advantages and ultimately contributes to the achievement of RTI. This study offers guidance for research teams to bolster their competitive advantages, thereby advancing the realization of RTI. The mediating effects of IL, IM, and RD&II are further scrutinized, with RD&II being deemed the most significant predictor of RTI mediation, as determined through sensitivity analysis utilizing ANN.

### 6.1 Theoretical implications

In accordance with the global trend of pursuing high-quality economic development led by science, technology, and innovation, this study provides the following theoretical implications. This study builds upon two foundational theories: the Innovation Drive Theory and the AMO Theory. Both theories scrutinize the interplay between social, organizational, and personnel behaviors and their implications on innovation. Drawing upon the assumptions of the AMO Theory, the empirical findings uncover that IPPs bolster team members' security, incentives, and opportunities in driving the realization of RTI.

This study proposes a framework for future research on team management in national research projects, with a focus on the interaction of IPP, IL, IM, RD&II, and GP within the RTI of scientific research teams. Unlike single-perspective analyses in some studies, this paper adopts a multi-dimensional approach. By contrast, previous researchers' emphasis on incremental technological innovations is replaced by a study of RTI, thus enriching the literature on technological innovation. Current research primarily focuses on the consequences of integrating technological innovation with Industry 4.0 [177], along with the moderating effect of IPP [7]. Nonetheless, these studies overlook the mediating influence. This study enrichs the existing body of knowledge by emphasizing the mediating roles of IL, IM, and RD&II, as well as the moderating impact of GP.

This study thoroughly investigates the impact of IPP on innovation-driven growth and technological advancement within national research program teams. By analyzing these teams, we can help countries improve their scientific and technological capacity, thereby boosting their competitiveness in the global market. As previously acknowledged, the limited investigation into RTI within national research programs by IPP, combined with the lack of contextual evidence related to these programs, emphasizes the relevance of this research to the existing literature. Therefore, this study expands the exploration of IPP, RTI in national research programs, IL, IM, RD&II, and GP.

### 6.2 Policy and managerial implications

This study provides several policy implications for both governments and organizations. Initially, legal and regulatory frameworks should establish clear intellectual property policies to promote technology transfer and collaboration in scientific research outcomes, in line with a country's specific capabilities and objectives. The multifaceted importance of national scientific research projects goes beyond mere scientific innovation, affecting economic development, social progress, and national security. Thus, the development of a robust intellectual property policy is essential for protecting the legitimate interests of scientific research outcomes. Furthermore, the creation of policies that promote collaboration between scientific

research clusters and industry, as well as the advancement of technology transfer, should be given prominence. Governments and organizations can enhance the innovation leadership of leaders by implementing training and development programs and increasing financial support for scientific research projects, thus enhancing the likelihood of RTI within scientific research teams. Limitations and future research directions will be discussed in the subsequent section.

## 6.3 Limitations and future research avenues

The study is not without its shortcomings, and there still exist unexplored avenues that require further investigation. This study presents preliminary findings, which can be used as a reference for further exploration on the relationships between IPPs, RTIs, ILs, IMs, RD&IIs, and GPs. However, future research should delve into the aspects related to technological innovation transformation and gain a deeper understanding of various dimensions, including economic environment, political environment, organizational situation, etc. This study primarily focuses on national research projects carried out within universities and research institutes, in contrast to enterprises. The research outcomes from enterprises are more market-relevant and environmentally friendly. Consequently, there is a requirement to enhance the examination of enterprise research project teams. To verify the claims made in this paper, we propose further empirical research on the impact of IPPs on RTI. This analysis could be conducted by examining the influence of external factors, such as industry and market demand, regulatory mechanisms, economic conditions, and competitor activities. This study takes into account both internal factors, including quality control system and knowledge-sharing mechanism resource allocation, as well as external influences. Given a cross-sectional research design, we are unable to establish longitudinal connections between the studied structures. Consequently, upcoming research should utilize varied longitudinal study approaches to examine the lasting impact of IPR protection on RTI within national research project teams.

## Supporting information

**S1 Data.**
(XLSX)

**S2 Data.**
(XLSX)

## Author Contributions

**Conceptualization:** Wei Chen, Jianhui Yin, Ye Tian, Haixu Shang, Yuan Li.

**Data curation:** Jianhui Yin.

**Formal analysis:** Jianhui Yin, Ye Tian.

**Funding acquisition:** Wei Chen, Yuan Li.

**Investigation:** Wei Chen, Haixu Shang, Yuan Li.

**Methodology:** Jianhui Yin, Ye Tian, Yuan Li.

**Project administration:** Wei Chen, Jianhui Yin, Ye Tian.

**Resources:** Haixu Shang, Yuan Li.

**Software:** Jianhui Yin.

**Supervision:** Ye Tian, Haixu Shang.

**Validation:** Wei Chen, Ye Tian, Haixu Shang.

**Visualization:** Wei Chen, Haixu Shang, Yuan Li.

**Writing – original draft:** Jianhui Yin.

**Writing – review & editing:** Jianhui Yin.

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
