## [Decision Letter · Decision Letter 0]

17 May 2024

PONE-D-24-08554Study examining the significant role of intellectual property protection in driving radical technological innovation among national research project teams, employing PLS-SEM and ANN modeling.PLOS ONE

Dear Dr. Yin,

Thank you for submitting your manuscript to PLOS ONE. After careful consideration, we feel that it has merit but does not fully meet PLOS ONE’s publication criteria as it currently stands. Therefore, we invite you to submit a revised version of the manuscript that addresses the points raised during the review process.

**Please address all the comments. **

We look forward to receiving your revised manuscript.

Kind regards,

Kashif Ali, PH.D

Academic Editor

PLOS ONE

Journal Requirements:

This work was supported in part by the Key projects for economic and social development in Heilongjiang Province of China under Grant 23301, in part by the Harbin Science and Technology Bureau Science and Technology Programme Projects of China under Grant ZC2023ZJ014007, and in part by the Heilongjiang Oriental College Research and Innovation Team Building Project of China under Grant HDFKYTD202108.

3. In the online submission form, you indicated that Data cannot be shared publicly because of the data is from team experiments. Data are available from the East University of Heilongjiang Access(contact via Jianhui Yin) for researchers who meet the criteria for access to confidential data.

Reviewers' comments:

Reviewer's Responses to Questions

**Comments to the Author**

1. Is the manuscript technically sound, and do the data support the conclusions?

Reviewer #1: Yes

Reviewer #2: Yes

Reviewer #3: Yes

Reviewer #4: Partly

2. Has the statistical analysis been performed appropriately and rigorously? 

Reviewer #1: Yes

Reviewer #2: Yes

Reviewer #3: Yes

Reviewer #4: I Don't Know

3. Have the authors made all data underlying the findings in their manuscript fully available?

Reviewer #1: Yes

Reviewer #2: Yes

Reviewer #3: Yes

Reviewer #4: Yes

4. Is the manuscript presented in an intelligible fashion and written in standard English?

Reviewer #1: Yes

Reviewer #2: Yes

Reviewer #3: Yes

Reviewer #4: Yes

5. Review Comments to the Author

**Reviewer #1**: The manuscript is well structured, it is suggested that kindly run grammarly for checking any typos and errors. Also, try to enhance the language of paper by using academically mature words so that the manuscript shall look sound enough to the readers.

**Reviewer #2:** I am delighted to review the article entitled "Study examining the significant role of intellectual property protection in driving radical technological innovation among national research project teams, employing PLS-SEM and ANN modeling." Authors have chosen a very interesting topic, but there are some areas which need to be corrected.

1. There are numerous repeated abbreviations and annotations in the text that require modification. When the words first appear, you can abbreviate the remarks, and you can use abbreviations directly afterwards.

2. The authors need to provide a more detailed literature review of the study, such as a literature review on national research project teams in driving innovation-driven growth in IPP and technological progress in previous studies. Other literature reviews should also include IL, IM, RD&II, and GP, but the assumptions section only includes a few. Relevant literature needs to be presented in the table form.

3. In the discussion, considering the different methods of PLS-SEM and ANN, are the key predictive factors obtained consistent or inconsistent? The discussion did not include the comparison results of the two methods.

4. Solving CMB with marker variables allows for a comparison of path efficiency (β) and p value. You could include a description in this section to see if the changes in the path coefficient and p-value are within 10%.

**Reviewer #3**: An interesting topic but few aspects need to be revisit.

-the specific research question and objectives need to be improved but make it simple

-need elaboration on how the underpinning theories help to explain the phenomena of the study/gap.

-further explanations are required why ANN analysis are needed? how it help to achieve the research gaps?

-Lack of measurement details and references

-lack of critical discussion in the discussion part.

**Reviewer #4: **Although the original data has been provided, it was written in Mandarin language. Perhaps, it is best to have an English version too, considering English as an international language and widely spoken among scholars/researchers.

6. PLOS authors have the option to publish the peer review history of their article (what does this mean?). If published, this will include your full peer review and any attached files.

Reviewer #1: No

Reviewer #2: No

Reviewer #3: **Yes: **sharizal hashim

Reviewer #4: No

---

## [Author Response · Author response to Decision Letter 0]

6 Jun 2024

We are grateful for the opportunity to submit a revised draft of the manuscript "Study examining the significant role of intellectual property protection in driving radical technological innovation among national research project teams, employing PLS-SEM and ANN modeling" for publication in the PLOS ONE. We appreciate the time and effort that editors and the reviewers dedicated to providing feedback on our manuscript and are grateful for the insightful comments and valuable improvements to our paper. We have incorporated most of the suggestions made by the reviewers. Please see "Response to Reviewers", for a point-by-point response to the re- viewer’ comments and concerns. All section numbers refer to the revised manuscript file with tracked changes.

---

## [Decision Letter · Decision Letter 1]

28 Jun 2024

Study examining the significant role of intellectual property protection in driving radical technological innovation among national research project teams, employing PLS-SEM and ANN modeling

PONE-D-24-08554R1

Dear Dr. Yin,

We’re pleased to inform you that your manuscript has been judged scientifically suitable for publication and will be formally accepted for publication once it meets all outstanding technical requirements.

Kind regards,

Kashif Ali, PH.D

Academic Editor

PLOS ONE

Additional Editor Comments (optional):

Reviewers' comments:

Reviewer's Responses to Questions

**Comments to the Author**

1. If the authors have adequately addressed your comments raised in a previous round of review and you feel that this manuscript is now acceptable for publication, you may indicate that here to bypass the “Comments to the Author” section, enter your conflict of interest statement in the “Confidential to Editor” section, and submit your "Accept" recommendation.

Reviewer #2: All comments have been addressed

Reviewer #3: All comments have been addressed

2. Is the manuscript technically sound, and do the data support the conclusions?

Reviewer #2: Yes

Reviewer #3: Yes

3. Has the statistical analysis been performed appropriately and rigorously? 

Reviewer #2: Yes

Reviewer #3: Yes

4. Have the authors made all data underlying the findings in their manuscript fully available?

Reviewer #2: Yes

Reviewer #3: Yes

5. Is the manuscript presented in an intelligible fashion and written in standard English?

Reviewer #2: Yes

Reviewer #3: Yes

6. Review Comments to the Author

Reviewer #2: (No Response)

Reviewer #3: Congratulation because able to address the reviewer comments. Suggest to do professional proofreading because some part are to lengthy.

7. PLOS authors have the option to publish the peer review history of their article (what does this mean?). If published, this will include your full peer review and any attached files.

Reviewer #2: No

Reviewer #3: **Yes: **sharizal hashim

---

## [Editor Report · Acceptance letter]

1 Jul 2024

PONE-D-24-08554R1 

PLOS ONE

Dear Dr. Yin, 

I'm pleased to inform you that your manuscript has been deemed suitable for publication in PLOS ONE. Congratulations! Your manuscript is now being handed over to our production team.

Kind regards, 

on behalf of

Dr. Kashif Ali 

Academic Editor

PLOS ONE